# Vitamin E-Inhibited Phoxim-Induced Renal Oxidative Stress and Mitochondrial Apoptosis In Vivo and In Vitro of Piglets

**DOI:** 10.3390/antiox12112000

**Published:** 2023-11-14

**Authors:** Jing Zhang, Yuecheng Sun, Wentao Song, Anshan Shan

**Affiliations:** Institute of Animal Nutrition, Northeast Agricultural University, Harbin 150030, China; jingzhang@fafu.edu.cn (J.Z.); sunyc1992@icloud.com (Y.S.); 18003363461@163.com (W.S.)

**Keywords:** phoxim, vitamin E, oxidative stress, mitochondrion, apoptosis

## Abstract

Exposure to phoxim at low levels caused bioaccumulation with neurotoxicity but also induced oxidative stress, tissue damage, and abnormal nutrient metabolism. This study described that vitamin E ameliorates phoxim-induced nephrotoxicity via inhibiting mitochondrial apoptosis. In vivo, 24 healthy piglets were treated with phoxim (0 mg/kg and 500 mg/kg) and vitamin E + phoxim (vitamin E + phoxim: 200 mg/kg + 500 mg/kg). In vitro, PK15 cells were treated with phoxim (0 mg/L and 1 mg/L) and vitamin E + phoxim (phoxim + vitamin E: 1 mg/L + 1 mg/L) for 12 h and 24 h. Our results indicated that accumulation of ROS, oxidative stress, and renal cell injury through stimulation of mitochondrial apoptosis resulted in phoxim-induced nephrotoxicity. Phoxim resulted in swollen mitochondria, blurred internal cristae, renal glomerular atrophy, and renal interstitial fibrosis. Vitamin E alleviated the adverse effects of phoxim by reducing ROS and improving antioxidant capacity in vivo and in vitro. Vitamin E significantly increased SDH in vitro (*p* < 0.01), while it decreased ROS, Bad, and cyto-c in vitro and SOD and CAT in vivo (*p* < 0.05). Vitamin E ameliorated phoxim-induced renal histopathologic changes, and mitochondria swelled. In addition, vitamin E regulates phoxim-induced apoptosis by alleviating oxidative damage to the mitochondria.

## 1. Introduction

Phoxim, an organophosphorus pesticide (OP), is frequently utilized in agricultural and veterinary fields, resulting in environmental pollution and food safety upon flowing into the food chain [1]. Phoxim accumulated in the respiratory tract and gastrointestinal tract is absorbed into the blood and circulated to target organs to exert its toxic effects in animals. Phoxim not only exerts its neurotoxicity through the cholinergic system but also induces oxidative stress, tissue damage, and abnormal nutrient metabolism [2]. It was found that chronic exposure to low doses of phoxim did not induce cholinergic neurotoxicity [3] but instead promoted the accumulation of reactive oxygen species (ROS) [4]. Phoxim not only inhibits cholinesterase activity but also induces a large amount of ROS [5]. Jin et al. found that ROS is an important apoptotic signal [6]. As a free radical scavenger, vitamin E (α-tocopherol) prevents the spread of free radical reactions by scavenging peroxy and alkoxy intermediates, and it protects cell membranes, mitochondria, and lipoproteins from oxidative damage [7]. Vitamin E has also been shown to regulate apoptotic regulators such as Bcl-2 and caspase-3 [8]. Recent studies have found that vitamin E alleviates the toxicity that is induced by exogenous pollutants in vivo and in vitro [9,10]. Mitochondria is the regulatory center of cells, plays a key role in regulating apoptosis, and is vulnerable to attack by heterogenous substances [11]. Mitochondria not only induce apoptosis but also enlarge the apoptotic signal. Four complexes form the mitochondrial respiratory chain on the mitochondrial inner membrane, including NADH dehydrogenase, succinate dehydrogenase, CoQH_2_-cytochrome C reductase, and cytochrome C oxidase. In the normal physiological state, the NADH dehydrogenase and CoQH_2_-cytochrome C reductase in the mitochondria produce a small amount of ROS when the mitochondrial respiratory chain transmits electrons [12]. The ROS is cleared by SOD in the mitochondria to maintain intracellular balance. A large number of ROS accumulate in cells under oxidative stress and lead to mitochondrial dysfunction. Mitochondrial dysfunction includes the leaking of electrons when the mitochondrial respiratory chain transmits electrons [13]. The leaked electrons react with oxygen to produce more ROS, which aggravates the vicious circle of mitochondrial dysfunction [14]. Van found that the accumulation of ROS could induce mitochondrial damage through mechanisms such as changing the membrane potential and structure of mitochondria and increasing the total amount of electron leakage and ROS production [15]. ROS has been described as an inducer of renal dysfunction [16]. The kidney is a key target organ that is involved in metabolism and has immunological and vital metabolic functions. The kidney has the highest mitochondrial number and oxygen consumption [17]. Previous research found phoxim-induced intestinal toxicity and hepatotoxicity, and vitamin E minimized intestinal and hepatic oxidative stress damage significantly [2,5]. At present, knowledge of the mechanism by which vitamin E alleviates phoxim-induced toxicity is rare. Based on previous research, this experiment was to evaluate vitamin E-regulated phoxim-induced renal oxidative stress injury and apoptosis by alleviating mitochondrial oxidative damage in vivo and in vitro.

## 2. Materials and Methods

### 2.1. Chemicals

In vivo experiments, phoxim (purity ≥ 92%) was supplied by Hubei Dixin Chemical Manufacturing Co., Ltd. (Wuhan, China). Vitamin E (α-tocopherol, purity ≥ 50%) was purchased from the Vitamin Factory of Zhejiang Medicine Co., Ltd. (Hangzhou, China). In vitro experiments, phoxim and vitamin E (α-tocopherol) were purchased from Sigma (St. Louis, MO, USA).

### 2.2. Experimental Design

#### 2.2.1. Animal Experiments

Twenty-four healthy 5-week-old male piglets of similar weights (16.5 ± 1.5 kg) were divided into 3 groups, including the control group, phoxim group (phoxim: 500 mg/kg), and vitamin E + phoxim group (vitamin E + phoxim: 200 mg/kg + 500 mg/kg), respectively. The sample size determined by the piglets’ model experiment is mainly based on the previous article [18]. After one week of pre-feeding, all the piglets were fed their corresponding diet for 30 consecutive days. Phoxim and vitamin E were dissolved in soybean oil immediately before use. The control group was given the same soybean oil. The composition and nutrient levels of the basal diet are presented in Table 1. All the piglets were fed the basal diet and were maintained according to the National Research Council Guide (1996) in metabolic cages. This study was approved by the Ethical and Animal Welfare Committee of Heilongjiang Province, China (2008). Animal care and treatment were in strict accordance with the standards of the Experimental Animal Care and Use Guide of the Northeast Agricultural University (NEAU-(2011)-9).

#### 2.2.2. Cell Experiments

PK15 cells were treated with phoxim (0 mg/L and 1 mg/L) and vitamin E + phoxim (phoxim + vitamin E: 1 mg/L + 1 mg/L). Eight replicates were used for each group, and the groups were incubated for 12 h and 24 h.

#### 2.2.3. Sample Collection

Piglets were fasted for 12 h and then sacrificed on the 30th day of the experiment. The tissues were immediately separated, frozen in liquid nitrogen, and stored in a cryogenic refrigerator at −80 °C. The remaining samples were stored at −20 °C. The blood was centrifuged for 15 min at 3000× *g* at 4 °C, and the serum samples were stored at −20 °C.

#### 2.2.4. Serum Biochemical Analysis

In vivo, the serum creatinine (CRE) and blood urea nitrogen (BUN) levels of piglets were tested using an automatic biochemical analyzer (Fully, Italy).

#### 2.2.5. Oxidation–Antioxidant Parameters

In vivo, the serum ROS levels were determined in the piglets by an enzyme-linked immunosorbent assay (ELISA), which was operated strictly according to the instructions of the kit (Shanghai Jinma Co., Ltd., Shanghai, China) on a Labsystems Multiskan MS (Finland). The antioxidant enzyme parameters were tested using an ultraviolet spectrophotometer (UV-2410PC model, Shimadzu Corp., Kyoto, Japan) according to the instructions of the commercial diagnostics kit (Nanjing Jiancheng Biotechnology Co., Ltd., Nanjing, China). The parameters that were evaluated in the kidney were malondialdehyde (MDA), total antioxidant capacity (T-AOC), superoxide dismutase (SOD), glutathione (GSH), catalase (CAT), and glutathione peroxidase (GSH-Px).

In vitro, the oxidation–antioxidant were ROS, SOD activity, and MDA content were determined according to the manufacturer’s instructions. The protein concentration was determined by a BCA protein detection kit. A model 680 enzyme labeling instrument (Bio-Rad, Hercules, CA, USA) was used for the detection.

#### 2.2.6. Histopathological Examinations

The kidney samples of piglets were fixed in 10% buffered formalin. Fixed samples were dehydrated through graded alcohols hyalinized by xylene and then embedded in paraffin blocks. Stained kidney sections with hematoxylin–eosin (H&E) at 5 μm. The sections were observed under an optical microscope.

#### 2.2.7. Ultrastructure of the Mitochondria

In vivo, the kidneys of piglets were collected, and 2.5% glutaraldehyde was used to fix the tissues to avoid light. In vitro, the PK15 cells were collected at 1000× *g* for 5 min, and then 2.5% glutaraldehyde was fixed in the cells. After they were fixed, the tissues were washed 4 times (15 min/wash) with 4 °C PBS (0.1 M, pH 7.2). The samples were immersed in 1% osmium tetroxide for 1 h (4 °C) and dehydrated in different concentrations of ethanol (50%, 70%, 90%, and 100%). After their dehydration, the samples were immersed in a mixture of acetone and acetone and embedded in epoxy resin. Finally, the samples were stained with uranium acetate and lead citrate and cut into ultrathin sections for observation under transmission electron microscopy.

#### 2.2.8. Mitochondrial Function

For mitochondrial function, the activities of succinate dehydrogenase (SDH), cytochrome c oxidase (COX), and Ca^2+^-Mg^2+^-ATPase in the PK15 cells were determined by ELISA. The enzyme activity was detected by a Labsystems Multiskan MS (Finland) system in strict accordance with the instructions of the kit (Shanghai Jinma Co., Ltd., Shanghai, China).

#### 2.2.9. Flow Cytometer Analysis of Apoptosis

In vitro, 1–5 × 10^5^ cells were collected and centrifuged at 1000× *g* for 5 min. Then, 195 L Annexin V-FITC was added to the cell precipitate, and 5 L Annexin V-FITC was added after mixing, which was then followed by the addition of a 10 L propidium iodide staining solution. After incubation at room temperature in the dark for 15 min, the samples were filtered through a 200-mesh filter (before adding PI) and analyzed by flow cytometry (Becton-Dickinson, San Jose, CA, USA) within 1 h. The excitation wavelength was 488 nm, and the emission wavelength was 530 nm. The green fluorescence of Annexin V-FITC was detected by FITC (FL-1, 530 nm), and the red fluorescence of the PI was detected by PI channel (FL-2, 585 nm).

#### 2.2.10. Mitochondrial Membrane Potential

1 mL of JC-1 dye solution was added to the PK15 cells and mixed well for 20 min in a carbon dioxide incubator (Thermo, Waltham, MA, USA). The 5X JC-1 dye buffer was diluted to 1X and placed on ice for 20 min. The cells were washed with the JC-1 staining buffer (1X), and the instructions of the kit were followed (JC-1 Biyun Tian, Shanghai, China).

#### 2.2.11. Quantitative Real-Time PCR (qRT-PCR)

Total RNA in tissues and cells was isolated using the TRIzol reagent kit, which was obtained from TaKaRa^®^ Bio Catalog (Dalian, China). All procedures were performed following the manufacturer’s instructions. The expression of genes, including GAPDH (NM_001206359.1; F: ATGCTTCTAGGCGGACTGT; R: CCATCCAACCGACTGCT), Bcl-2 (EF681866.1; F: CATGTGTGTGGAGAGCGTCA; R: GCATCCCAGCCTCCGTTATC), Bad (XM_005660745; F: TGAAGGGACTGAGGATGAGG; R: GAAGGAACCCTGGAACTCGT), Bax (XM_005664710; F: ATGGAGCTGCAGAGGATGAT; R: AAAGTAGAAAAGCGCGACCA), Caspase-3 (NM_214131; F: CGGACAGTGGGACTGAAGAT; R: GATCCGTCCTTTGAATTTCG), Caspase-8 (NM_001031779; F: TTGGGGAACATTTGGACAGT; R: TTTTCTTGGAGCCTCTGGAA), Caspase-9 (XM_003127618.3; F: TGAACTTCTGCCATGAGTCG, R: ATTTGCTTGGCAGTCAGGTT), Cyto-C (NM_001129970; F: AAAGGGAGGCAAACACAAGA; R: CCAGGTGATGCCTTTGTTCT), FAS (NM_213839; F: CCACGTGTGAACATGGAGTC; R: GAGGGCCCATAACCAGTGTA) were determined by Sangon Biological Engineering Co., Ltd. (Shanghai, China) and analyzed on the ABI PRISM 7500 SDS thermal cycler apparatus (Applied Biosystems, Foster City, CA, USA). The expression level of genes was measured by the 2^−ΔΔCt^ method.

### 2.3. Statistical Analysis

All data were analyzed using an analysis of variance (ANOVA) test and the least significant difference (LSD) procedure in SPSS (version 22.0; IBM-SPSS Inc., Chicago, IL, USA). Duncan’s multiple range test determined the differences between groups. All the data were presented as the mean ± SEM. *p* < 0.05 was a significant difference and was denoted by different lowercase letters, while with different capital letters, it means *p* < 0.01.

## 3. Results

### 3.1. Serum Biochemical Analysis

Regarding serum biochemical parameters, they are shown in Table 2. For renal function, phoxim induction significantly increased the content of serum BUN (*p* < 0.05) and CRE (*p* < 0.01). The supplementation of 200 mg/kg vitamin E significantly suppressed the phoxim-induced increase of the CRE (*p* < 0.01).

### 3.2. Oxidant-Antioxidant Parameters

Data on oxidative-antioxidant parameters are shown in Table 3 and Table 4. In vivo (Table 3), phoxim significantly increased serum ROS and MDA (*p* < 0.01), while it significantly decreased SOD, T-AOC, GPx, CAT, and GSH (*p* < 0.05). Compared with the phoxim group, SOD and CAT were significantly decreased in the kidneys of the piglets in the vitamin E + phoxim group (*p* < 0.01).

In vitro (Table 4), phoxim increased ROS and MDA significantly in 12 and 24 h (*p* < 0.05, *p* < 0.01), while SOD was decreased by phoxim induction (*p* < 0.05). Compared with the phoxim group, vitamin E decreased ROS significantly in 24 h (*p* < 0.05). Vitamin E declined the phoxim induction elevated in ROS, and there was no significant difference in ROS and SOD between the control group and the vitamin E + phoxim group (*p* > 0.05).

### 3.3. Histopathological Findings

Histopathologic sections of piglets’ kidneys were observed in Figure 1. (a) The control group showed the normal appearance of renal glomeruli (thick arrow) and renal tubule (thin arrow). (b) Phoxim-exposed piglets showed renal glomerular atrophy (thick arrow) and renal interstitial fibrosis (thin arrow). (c) The group of vitamin E + phoxim showed normal glomeruli (thick arrow) and tubules in the kidney (thin arrow).

### 3.4. Ultrastructure of the Mitochondria

Transmission electron microscopy of piglets’ kidneys and pk15 cells is shown in Figure 2 and Figure 3.

In vivo (Figure 2), (a) the Control group showed the normal appearance of nephrocytes. (b) After exposure to phoxim, the mitochondria gradually become swollen, and the number of mitochondria decreases. (c) The mitochondria were slightly swollen in the vitamin E + phoxim group.

In vitro, phoxim induced the mitochondria to swell, the internal crest blurred and fractured, and a small amount of vacuolation appeared for 12 h. For 24 h, phoxim induced the deformation and disappearance of mitochondrial membranes. There were a large number of vacuoles that appeared in mitochondria with the addition of phoxim for 24 h. The mitochondria were slightly swollen in the vitamin E + phoxim group.

### 3.5. Mitochondrial Function

As for mitochondrial function enzymes (Table 5), phoxim induction significantly declined COX (*p* < 0.05), Ca^2+^-Mg^2+^-ATPase, and SDH (*p* < 0.01) for 24 h. Vitamin E elevated the phoxim-inducted decline in SDH (*p* < 0.01), COX, and Ca^2+^-Mg^2+^-ATPase. There was no significant difference in COX and SDH activity between the control and vitamin E + phoxim (*p* > 0.05).

### 3.6. Cell Apoptosis Assay

Regarding cell apoptosis rate (Table 6 and Figure 4), phoxim significantly increased the apoptotic rate of the PK15 cells for 24 h (*p* < 0.01), and there was no significant difference between the control group and vitamin E + phoxim group (*p* > 0.05).

### 3.7. Mitochondrial Membrane Potential

For mitochondrial membrane potential (Table 7), the decrease (*p* < 0.05) caused by phoxim induction in the mitochondrial membrane potential relative to the control was reduced by vitamin E, and there was no significant difference between the vitamin E + phoxim group and the control group.

### 3.8. Quantitative Real-Time PCR (qRT-PCR)

As for the mRNA expressions of apoptosis signaling factors (Table 8 and Table 9), phoxim significantly elevated the expression of Bad, Bax, caspase-3, and caspase-9 in vivo and in vitro (*p* < 0.05), while the expression of Bcl-2 was firmly lower than the control group (*p* < 0.01). There was no significant difference in Bad, Bax, caspase-3, Bcl-2, and caspase-9 between the control group and the vitamin E + phoxim group (*p* > 0.05). In vitro, phoxim increased the mRNA expression of cyto-c (*p*< 0.05) relative to the phoxim group, which was reduced (*p*< 0.05) by vitamin E.

## 4. Discussion

Phoxim is widely utilized in agriculture and veterinary fields and accumulates in tissues to cause adverse effects on farm animals and human health. The toxic mechanism is related to impaired oxidative homeostasis, including the overproduction of ROS and oxidative stress [18,19,20]. As an effective lipid-soluble antioxidant, vitamin E alleviates the toxic effects of many organophosphorus pesticides in vivo and in vitro [9,10].

### 4.1. Serum Biochemical Analysis

The content of BUN and CRE in serum was an evaluation of renal function, which accurately reflected the renal injury. BUN is related to the glomerular filtration function, protein absorption, and urea excretion rate [21]. CRE is a product of muscle metabolism, which depends on the ability of glomerular filtration [22]. The overproduction of ROS impaired renal function with increasing serum BUN and CRE [16]. It is well known that vitamin E regulates oxidative stress and creatine metabolism and improves protein utilization. Our study found that phoxim significantly increased the content of BUN and CRE in the serum of piglets, indicating that phoxim-induced nephrotoxicity, which is attributed to renal oxidative damage, is well in line with the histological changes. The supplementation of 200 mg/kg vitamin E suppressed the phoxim-induced increase in serum CRE significantly, while serum BUN increased insignificantly. In addition to phoxim, these effects might be related to the dosage and duration of vitamin E, and the mechanism of action needs further study.

### 4.2. Oxidation-Antioxidation

In an environmental risk assessment, oxidation-induced changes are considered biomarkers of environmental pollution [23]. The oxidative process and accumulation of ROS in cells cause mitochondrial dysfunction [15], which in turn alters the mitochondrial membrane potential and structure, increases the total amount of ROS, and forms a vicious cycle. Studies reported that Ops-induced increases in ROS led to significant changes in mitochondrial transmembrane potential and resulted in mitochondrial dysfunction under oxidative stress [24]. The toxic mechanism of OPs is related to the overproduction of ROS [25]. MDA is a metabolite of lipid peroxidation and is generally considered to be the most representative end product of oxidative stress [26,27]. It is used to evaluate lipid peroxidation and oxidative damage. Our results of the animal and cell experiments showed that phoxim significantly increased ROS and MDA, which means oxidative stress, which is consistent with other findings [28,29,30]. The antioxidant enzyme system (SOD, GPX, GST, and CAT) and the non-enzymatic antioxidant system (vitamin E, vitamin C, and GSH) consist of an antioxidant system that protects the body from oxidative stress damage [31,32]. SOD is a first-line defense mechanism that converts ROS into hydrogen peroxide [33]. GPx and CAT are the second lines of the antioxidant defense mechanism, which act by converting hydrogen peroxide into hydrogen peroxide and oxygen peroxide [34]. As a non-enzymatic antioxidant, GSH is a tripeptide that contains cysteine, which directly scavenges ROS, participates in the GPx catalytic reaction, and protects the body from lipid peroxidation toxicity [10,26]. GSH prevents the release of cytochrome C from the mitochondria. We found that phoxim-induced excessive ROS resulted in abnormal lipid peroxidation, increased MDA, and decreased SOD in vivo and in vitro. Phoxim significantly reduced GPx, T-AOC, GSH, and CAT in the kidneys of piglets, and these findings are similar to those of previous reports [25,32,35]. The results showed that phoxim induced lipid peroxidation and oxidative stress. As a member of the non-enzymatic antioxidant system, vitamin E inhibited the accumulation of ROS by health-related physiological processes and alleviated oxidative stress [36,37,38]. Many studies have suggested that vitamin E supplementation is a potential strategy for improving oxidative stress damage [39,40]. Our study found that vitamin E alleviates OP-induced oxidative stress by lowering the ROS and increasing the CAT in vivo and in vitro, which were similar to previous reports [33,41,42]. In summary, phoxim impaired oxidative homeostasis through the accumulation of ROS and MDA while decreasing SOD, GPx, GSH, and CAT in vivo and in vitro. Vitamin E scavenged ROS by increasing SOD and improved the antioxidant capacity by elevating GPx, GSH, and CAT, which alleviated phoxim-induced mitochondrial oxidative stress injury and led to hepatotoxicity.

### 4.3. Histopathological Findings

Histopathological examination is not only a standard method for assessing the degree of tissue damage but is also used as an index to evaluate the effect of exogenous pollutants [43]. The kidney is an important detoxification and excretory organ in animals. The results of the present study showed renal tubular epithelium and glomerulus lumen swelling, vacuolization, and vacuolization-necrosis in OPs-treated animals’ kidneys [44,45]. Our study found that phoxim-exposed piglets showed injuries in the kidney structure, such as renal glomerular atrophy and renal interstitial fibrosis. Vitamin E ameliorated phoxim-induced renal histopathologic changes, showing normal glomeruli and tubules in the kidney. Also, histological findings supported the results of serum biochemical analysis, which were similar to previous studies [44,46].

### 4.4. Ultrastructure of the Mitochondria

As an active metabolic organ, the kidney has plenty of mitochondria, which regulate cellular metabolism and apoptosis [16,47]. Intracellular ROS are mainly derived from mitochondria [48]. ROS overproduction causes mitochondrial oxidative damage and kidney epithelial cell apoptosis, which results in mitochondrial fragmentation [49]. It is reported that mitochondrial fragmentation may accelerate kidney epithelial cell apoptosis and inhibit kidney recovery from kidney injury [50]. Dirican and Kalender found that subacute and subchronic dichlorvos exposure results in vacuolation and swelling of mitochondria [51]. Li found that as the trichlorfon dosage increases, the mitochondria become blurred, and the degree of mitochondrial swelling gradually increases until the mitochondria rupture or defect [52]. We demonstrated phoxim-induced nuclear atrophy and mitochondrial swelling, which were the typical apoptotic characteristics. Phoxim caused the mitochondria of the PK15 cells to gradually become swollen, a large degree of vacuolation appeared, and the internal cristae became blurred; this deformation seriously altered the mitochondria and eventually caused them to disintegrate, which caused the number of mitochondria to decrease. Cell damage triggered the mitochondrial pathways to participate in the caspase-dependent apoptotic response [53]. Vitamin E treatment alleviated phoxim-induced mitochondrial changes and noted renoprotection. 

### 4.5. Mitochondrial Enzyme Activity

The mitochondria are the energy centers of cells [54]. The mitochondrial function includes cell energy metabolism, electron transport in the respiratory chain, and Ca^2+^ storage, which is closely related to cell metabolism and determines the survival of cells. Ca^2+^-Mg^2+^-ATPase, cytochrome c oxidase (COX), and succinate dehydrogenase (SDH) are three enzymes that reflect mitochondrial function. Ca^2+^-Mg^2+^-ATPase is mainly distributed on the cell membrane and plays an important role in maintaining the integrity and fluidity of the cell membrane. The Ca^2+^-Mg^2+^-ATPase activity decreases when the cell membrane is damaged. OPs significantly reduced the activity of Ca^2+^-Mg^2+^-ATPase [55]. COX is most vulnerable to peroxides and plays an important role in regulating mitochondrial oxidation. The activity of COX could accurately reflect the function of mitochondria. Studies have shown that COX activity is inhibited when mitochondrial function is impaired. SDH is the key enzyme of the TCA cycle, which is embedded in the mitochondrial inner membrane. The activity of SDH reflects the extent of the TCA cycle and mitochondrial function. It provides electrons for the mitochondrial respiratory chain and is the junction between the respiratory electron transport chain and oxidized phosphoric acid. We demonstrated that phoxim-induced COX, Ca^2+^-Mg^2+^-ATPase, and SDH significantly decreased for 24 h. Vitamin E treatment increased phoxim-induced COX, Ca^2+^-Mg^2+^-ATPase, and SDH activities, which were similar to Xiao’s results [56]. The results showed that phoxim had adverse effects on mitochondrial function, and vitamin E alleviated phoxim-induced mitochondrial dysfunction, which is linked to oxidative stress inseparably [48].

### 4.6. Apoptosis

#### 4.6.1. Mitochondrial Transmembrane Potential

Mitochondrial membrane potential (ΔΨm) is the difference in ion concentration between the two sides of the mitochondrial inner membrane, which is reflected in mitochondrial injury and linked to mitochondrial function [57]. Mitochondrial dysfunction, ΔΨm, and cell energy were decreased, leading to cell apoptosis. The reduction of ΔΨm is the first step in the apoptotic process. Studies have shown that OP significantly reduced ΔΨm in a dose-dependent manner [52,58]. This study found that phoxim-induced ΔΨm decreased, reflecting that the mitochondrial pathway was an important pathway for OPs-induced apoptosis. Adding vitamin E could effectively inhibit this trend.

#### 4.6.2. Apoptosis Rate

Apoptosis is a programmed form of cell death that is controlled by genes, and it plays a key role in physiological and pathological processes. As a regulatory mechanism to maintain the dynamic balance of the body’s internal environment, cellular apoptosis removes redundant, aging, and even damaged cells in the body, which achieves the purpose of maintaining the normal physiological activities of the body [59]. Abnormal mitochondrial function induces apoptosis [60]. We found that phoxim increased the apoptosis rate of the PK15 cells significantly for 24 h. It means that phoxim-induced apoptosis occurs in the PK15 cells. Vitamin E supplementation effectively inhibits phoxim-induced apoptosis rate increases.

#### 4.6.3. Apoptotic Factor mRNA

ROS and oxidative stress are apoptotic signals that induce apoptosis by activating apoptotic factors [61]. There were pro-apoptosis factors (Fas, FasL, Bax, and Caspase3) and the anti-apoptosis factor (Bcl-2) to evaluate apoptosis [16]. Apoptosis involved intrinsic pathways (mitochondrial-mediated) and extrinsic pathways (Fas/FasL mediate), and Caspase 3 was the executor [62,63]. Mitochondrial apoptosis is related to the Bcl-2 family, caspase-3, and caspase-9. Cytochrome c (cyto-c) is an electron transporter in the mitochondrial respiratory chain. During apoptosis, mitochondrial cyto-c releases into the cytoplasm and triggers caspases [64]. Caspase-9 activated caspase-3 and led to mitochondrial apoptosis. The death signal receptor protein pathway is mediated by a variety of death receptor ligands, such as Fas, FasL. Overexpression of Bcl-2 prevents the decrease in the mitochondrial membrane potential, the release of cyt c, and the activation of caspases to regulate apoptosis, which causes Bax-induced changes. Caspase family proteins changed cell morphology and induced apoptosis. Different studies demonstrated that OP induced mitochondrial changes and oxidative stress and resulted in apoptosis [16,65]. In vitro and in vivo experiments, we found that phoxim significantly increased the expression of Bad, Bax, cyto-c, caspase-3, caspase-8, and caspase-9 but reduced Bcl-2 expression, which means the apoptotic pathway was activated by phoxim. Combined with the previous results, phoxim induced the accumulation of ROS and led to mitochondrial dysfunction. Mitochondria were important targets for the toxic effects of phoxim. Phoxim exposure increased mitochondrial membrane permeability, decreased ΔΨm, released apoptotic factors into the cytoplasm, activated caspase-3 to trigger the caspase cascade reaction, and induced mitochondrial apoptosis. Thus, phoxim-induced oxidative stress resulted in apoptosis, which was one of the toxic mechanisms. Vitamin E significantly decreased phoxim-induced expression of caspase-8, Bad, and cyto-c. As an antioxidant, Vitamin E prevents the generation of ROS, improves the antioxidant capacity, and alleviates mitochondrial apoptosis.

## 5. Conclusions

Mitochondrial irreversible damage and oxidative stress are crucial contributors to phoxim-induced nephrotoxicity. Phoxim-induced mitochondrial oxidative injury accelerates mitochondrial apoptosis. Vitamin E prevented the generation of ROS and improved the antioxidant capacity to alleviate the phoxim-induced nephrotoxicity. Vitamin E regulates phoxim-induced apoptosis by alleviating oxidative damage to the mitochondria.

## Figures and Tables

**Figure 1 antioxidants-12-02000-f001:**
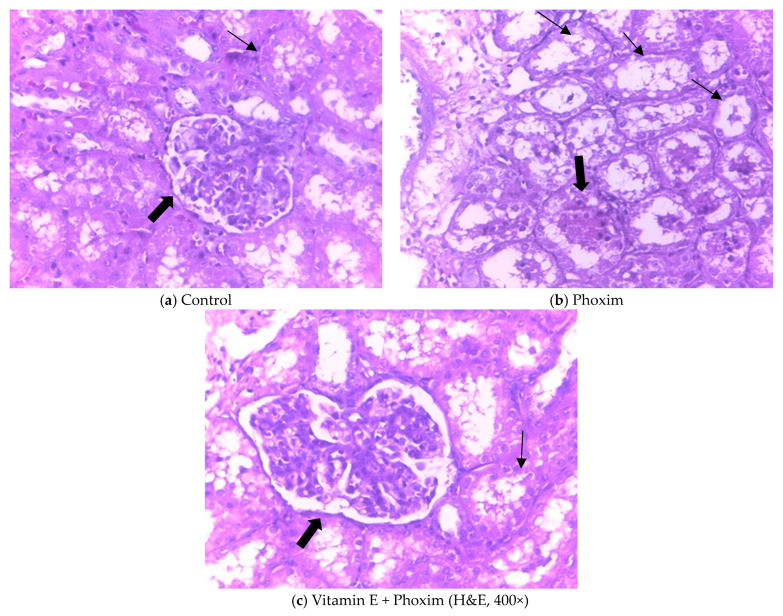
Histopathologic sections of piglets’ kidneys (H&E, 400×): In vivo, (**a**) Control group showed the normal appearance of renal glomeruli (thick arrow) and renal tubule (thin arrow). (**b**) Phoxim-exposed piglets showed renal glomerular atrophy (thick arrow) and renal interstitial fibrosis (thin arrow). (**c**) The group of vitamin E + phoxim showed normal glomeruli (thick arrow) and tubules in the kidney (thin arrow).

**Figure 2 antioxidants-12-02000-f002:**
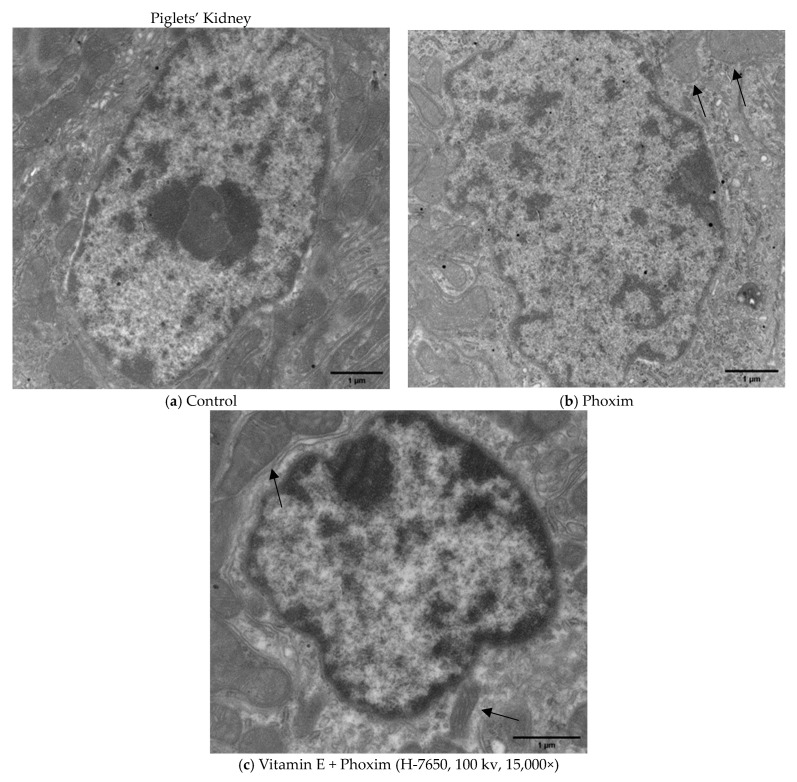
Transmission electron microscopy of piglets’ kidney (H-7650, 100 kv, 15,000×): In vivo, (**a**) Control group showed the normal appearance of nephrocytes. (**b**) After exposure to phoxim, the mitochondria gradually become swollen (thin arrow), and the number of mitochondria decreases. (**c**) The mitochondria were slightly swollen (thin arrow) in the vitamin E + phoxim group.

**Figure 3 antioxidants-12-02000-f003:**
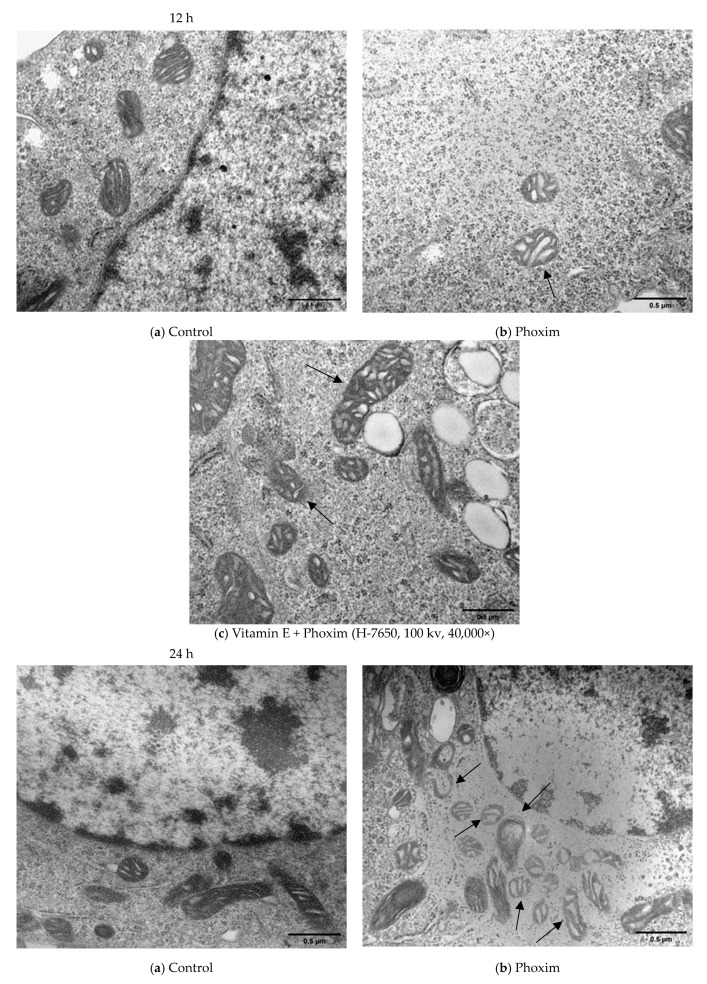
Transmission electron microscopy in vitro (H-7650, 100 kv, 40,000×): In vitro, (**a**) Control group showed the normal appearance of nephrocytes of 12 h and 24 h. (**b**) After exposure to phoxim, the mitochondria gradually become swollen (thin arrow), and the number of mitochondria decreases of 12 h and 24 h. (**c**) The mitochondria were slightly swollen (thin arrow) in the vitamin E + phoxim group of 12 h and 24 h.

**Figure 4 antioxidants-12-02000-f004:**
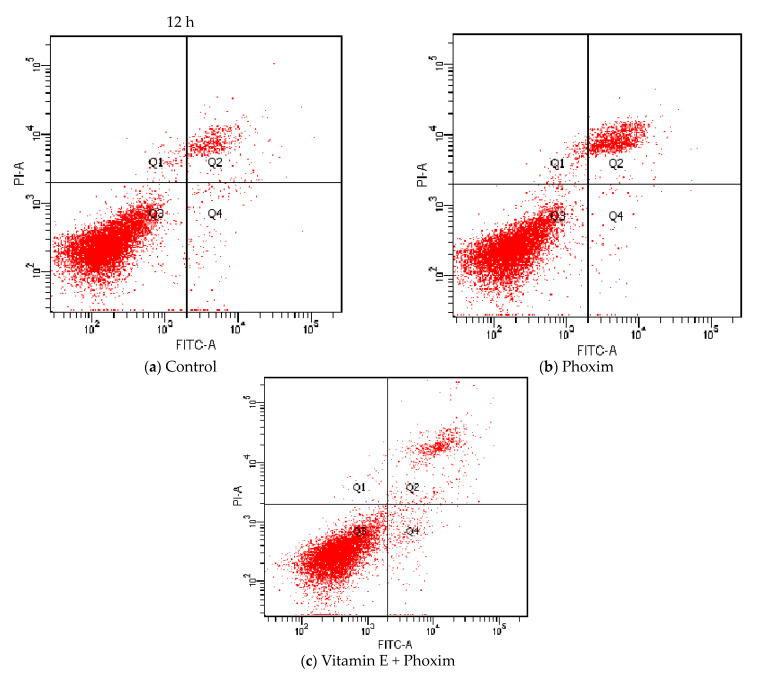
Vitamin E affects phoxim-induced cell apoptosis rate in vitro.

**Table 1 antioxidants-12-02000-t001:** Composition and nutrient levels of basal diet.

Components	(%)	Componsition ^3^	(%)
Corn	71.56	DE (MJ/kg)	14.12
Soybean meal	15.65	Crude protein	16.80
Fishmeal	3.00	Calcium	0.67
Soybean oil	2.00	Phosphorus	0.56
Wheat bran	5.00	Lysine	0.98
CaHPO_4_	0.80	Methionine	0.28
Limestone	0.78		
Salt	0.35		
Lysine	0.26		
Premix ^1^	0.50		
Choline chloride ^2^	0.10		
Total	100.00		

Note: ^1^ Provided the following per kilogram of diet: Cu 20 mg; Zn 80 mg; Se 0.2 mg; Mn 25 mg; Fe 100 mg; I 0.3 mg; vitamin A 8000 IU; vitamin D 2000 IU; vitamin E 30 IU; vitamin K_3_ 1.5 mg; vitamin B_1_ 1.6 mg; vitamin B_6_ 1.5 mg; vitamin B_12_ 12 μg; niacin 20 mg; D-pantothenic acid 15 mg. ^2^ The effective concentration of choline chloride is 50%. ^3^ Nutrient contents were calculated values.

**Table 2 antioxidants-12-02000-t002:** Vitamin E affects phoxim-induced renal function parameters in piglets.

Items	Control	Phoxim	Vitamin E + Phoxim
BUN (mmol/L)	2.73 ± 0.04 ^b^	3.68 ± 0.35 ^a^	3.17 ± 0.09 ^ab^
CRE (μmol/L)	92.08 ± 2.62 ^C^	121.82 ± 3.57 ^A^	106.20 ± 2.13 ^B^

Note: In the same row, different lowercase letter superscripts mean notable differences (*p* < 0.05), while different capital letters show significant differences (*p* < 0.01). Values with the same or no letters mean no significant difference (*p* > 0.05).

**Table 3 antioxidants-12-02000-t003:** Vitamin E affects phoxim-induced oxidative-antioxidant parameters in vivo.

Items	Control	Phoxim	Vitamin E + Phoxim
ROS in serum (IU/mL)	166.09 ± 8.69 ^C^	289.60 ± 3.84 ^A^	265.01 ± 1.35 ^B^
Kidney			
MDA (nmol/mg protein)	1.33 ± 0.09 ^B^	1.96 ± 0.16 ^A^	1.69 ± 0.08 ^AB^
SOD (U/mg protein)	254.39 ± 4.08 ^A^	203.06 ± 1.11 ^C^	228.06 ± 540 ^B^
T-AOC (U/mg protein)	0.43 ± 0.01 ^A^	0.31 ± 0.02 ^B^	0.36 ± 0.01 ^B^
GPx (nmol/mg protein)	31.18 ± 1.06 ^A^	21.8 ± 0.75 ^B^	26.10 ± 2.24 ^AB^
CAT (U/mg protein)	36.60 ± 2.78 ^A^	7.41 ± 1.55 ^C^	15.81 ± 0.79 ^B^
GSH (U/mg protein)	37.27 ± 3.36 ^A^	24.04 ± 1.10 ^B^	31.95 ± 1.60 ^AB^

Note: In the same row, different capital letters show significant differences (*p* < 0.01).

**Table 4 antioxidants-12-02000-t004:** Vitamin E affects phoxim-induced oxidative-antioxidant parameters in vitro.

Items	Control	Phoxim	Vitamin E + Phoxim
12 h			
ROS (%)	100.00 ± 5.70 ^B^	114.13 ± 2.72 ^A^	102.98 ± 2.05 ^B^
SOD (U/mgprot)	29.16 ± 0.22	25.26 ± 0.32	26.28 ± 1.87
MDA (μmol/mg)	9.82 ± 0.14 ^B^	11.38 ± 0.18 ^A^	11.02 ± 0.15 ^A^
24 h			
ROS (%)	100.00 ± 2.75 ^B^	121.83 ± 2.17 ^A^	107.62 ± 1.88 ^B^
SOD (U/mgprot)	26.81 ± 0.72 ^a^	21.94 ± 0.96 ^b^	23.42 ± 1.56 ^ab^
MDA(μmol/mg)	11.51 ± 0.1 ^B^	15.42 ± 0.59 ^A^	14.11 ± 0.48 ^A^

Note: In the same row, different lowercase letter superscripts mean notable differences (*p* < 0.05), while different capital letters show significant differences (*p* < 0.01). Values with the same or no letters mean no significant difference (*p* > 0.05).

**Table 5 antioxidants-12-02000-t005:** Vitamin E affects phoxim-induced mitochondrial function enzyme activity in vitro.

Items	Control	Phoxim	Vitamin E + Phoxim
12 h			
COX (IU/L)	360.24 ± 32.66	354.54 ± 7.51	326.82 ± 8.90
Ca^2+^-Mg^2+^-ATPase (ng/L)	338.34 ± 13.20	310.80 ± 11.90	361.36 ± 29.94
SDH (U/L)	597.06 ± 12.61	580.26 ± 27.89	589.03 ± 6.64
24 h			
COX (IU/L)	363.74 ± 14.00 ^a^	315.29 ± 14.09 ^b^	334.33 ± 3.76 ^ab^
Ca^2+^-Mg^2+^-ATPase (ng/L)	389.42 ± 4.09 ^A^	320.88 ± 9.41 ^B^	333.17 ± 16.59 ^AB^
SDH (U/L)	637.77 ± 3.83 ^A^	475.46 ± 11.54 ^B^	592.68 ± 18.03 ^A^

Note: In the same row, different lowercase letter superscripts mean notable differences (*p* < 0.05), while different capital letters show significant differences (*p* < 0.01). Values with the same or no letters mean no significant difference (*p* > 0.05).

**Table 6 antioxidants-12-02000-t006:** Vitamin E affects phoxim-induced cell apoptosis rate in vitro.

Items	Control	Phoxim	Vitamin E + Phoxim
12 h (%)			
Cells apoptosis rate	4.50 ± 0.12	8.16 ± 0.68	7.03 ± 1.70
24 h (%)			
Cells apoptosis rate	10.15 ± 0.49 ^B^	19.90 ±1.38 ^A^	15.24 ± 1.40 ^AB^

Note: In the same row, different capital letters show significant differences (*p* < 0.01). Values with the same or no letters mean no significant difference (*p* > 0.05).

**Table 7 antioxidants-12-02000-t007:** Vitamin E affects phoxim-induced mitochondrial membrane potential in vitro.

Items	Control	Phoxim	Vitamin E + Phoxim
12 h (%)			
Mitochondrial membrane potential	100.00 ± 5.18 ^a^	84.45 ± 3.02 ^b^	92.88 ± 3.97 ^ab^
24 h (%)			
Mitochondrial membrane potential	100.00 ± 5.44 ^A^	82.64 ± 2.44 ^B^	87.98 ± 4.20 ^AB^

Note: In the same row, different lowercase letter superscripts mean notable differences (*p* < 0.05), while different capital letters show significant differences (*p* < 0.01).

**Table 8 antioxidants-12-02000-t008:** Vitamin E affects phoxim-induced mRNA expressions of apoptosis factors in piglets.

Items	Control	Phoxim	Vitamin E + Phoxim
Bcl-2	1.25 ± 0.08 ^A^	0.74 ± 0.09 ^B^	1.19 ± 0.14 ^AB^
Bad	0.82 ± 0.12 ^b^	1.35 ± 0.16 ^a^	1.12 ± 0.11 ^ab^
Bax	0.65 ± 0.07 ^b^	1.18 ± 0.17 ^a^	1.00 ± 0.14 ^ab^
Caspase-3	0.78 ± 0.08 ^B^	1.22 ± 0.11 ^A^	1.02 ± 0.06 ^AB^
Caspase-8	0.66 ± 0.22 ^b^	1.32 ± 0.05 ^a^	0.92 ± 0.12 ^ab^
Caspase-9	0.80 ± 0.02 ^b^	1.34 ± 0.15 ^a^	1.32 ± 0.23 ^a^
Cyto-C	0.85 ± 0.08	1.16 ± 0.06	0.94 ± 0.16
FAS	1.13 ± 0.25	0.87 ± 0.09	0.85 ± 0.10

Note: In the same row, different lowercase letter superscripts mean notable differences (*p* < 0.05), while different capital letters show significant differences (*p* < 0.01). Values with the same or no letters mean no significant difference (*p* > 0.05).

**Table 9 antioxidants-12-02000-t009:** Vitamin E affects phoxim-induced mRNA expressions of apoptosis factors in PK15 cells.

Items	Control	Phoxim	Vitamin E + Phoxim
12 h			
Bad	0.79 ± 0.13 ^B^	1.64 ± 0.14 ^A^	1.09 ± 0.04 ^B^
Bax	0.74 ± 0.08 ^B^	1.13 ± 0.05 ^A^	0.98 ± 0.07 ^AB^
Bcl-2	1.38 ± 0.20 ^a^	0.81 ± 0.07 ^b^	1.07 ± 0.09 ^ab^
Caspase-3	0.72 ± 0.25 ^b^	1.40 ± 0.12 ^a^	1.18 ± 0.10 ^ab^
Caspase-8	1.32 ± 0.21	1.35 ± 0.17	1.02 ± 0.08
Caspase-9	0.76 ± 0.10 ^B^	1.48 ± 0.23 ^A^	1.06 ± 0.06 ^AB^
Cyto-C	0.86 ± 0.04 ^b^	1.16 ± 0.08 ^a^	0.92 ± 0.06 ^b^
24 h			
Bad	0.90 ± 0.07 ^b^	1.68 ± 0.47 ^a^	1.01 ± 0.06 ^b^
Bax	0.72 ± 0.10 ^b^	1.09 ± 0.02 ^a^	0.97 ± 0.09 ^ab^
Bcl-2	1.25 ± 0.05 ^A^	0.76 ± 0.12 ^B^	1.02 ± 0.05 ^AB^
Caspase-3	0.86 ± 0.40 ^b^	1.75 ± 0.07 ^a^	1.08 ± 0.22 ^ab^
Caspase-8	0.81 ± 0.27	1.82 ± 0.65	1.30 ± 0.23
Caspase-9	0.79 ± 0.12 ^B^	2.24 ± 1.00 ^A^	1.05 ± 0.10 ^AB^
Cyto-C	0.88 ± 0.10 ^B^	1.35 ± 0.10 ^A^	0.97 ± 0.07 ^B^

Note: In the same row, different lowercase letter superscripts mean notable differences (*p* < 0.05), while different capital letters show significant differences (*p* < 0.01). Values with the same or no letters mean no significant difference (*p* > 0.05).

## Data Availability

No new data were created or analyzed in this study. Data sharing is not applicable to this article.

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
