# Peer review of "Vitamin E-Inhibited Phoxim-Induced Renal Oxidative Stress and Mitochondrial Apoptosis In Vivo and In Vitro of Piglets"

_antioxidants, 2023, doi:10.3390/antiox12112000_

Round 1
Reviewer 1 Report
Major comments
L43-54: references are missing. Should be added to support the statements.
L189: the results section starts with report of results on biochemical parameters etc. It is suggested to the authors to modify the result section according to the materials and methods section, describing in subheading the results firstly of the in vivo results, and then the in vitro ones.
Tables 2, 3, 4..: a footnote in each table is missing regarding the superscripts explanation. Please note that in some tables lowercase is used, while in other rows an uppercase letter. This should be uniform. Also check that in some columns a superscript is missing, for example in table 3 ROS value for the phoxim group does not have a superscript. Finally, check those cases that do not differ between groups, should have the same superscript.
L336: “vitamin E significantly suppressed the phoxim-induced increase of the serum CRE and alleviated nephrotoxicity”: the authors should revise this sentence, as the vitamin E supplementation did not totally restore the CRE values to normal as compared to the control treatment. It should be further explained why these effects were noted in this case? Could this be relevant to the doses used or to the duration of exposure to the phoxim toxin.
L340-349: this section is rather a literature review, should be omitted or reduced to 1 sentence related to the content of the findings. Same remark for lines 352-359.
L368: the authors should check the interpretation of their findings again. For example in table 3, the MDA in the VitE group was not statistically significant compared to the other 2 groups. Then the statement “Our study found that vitamin E alleviates OP-induced oxidative stress 368 by lowering the ROS, and MDA levels” needs revision. The same remark is for ROS levels, and a superscript is missing from phoxim group in table 3.
Minor comments
L14: piglets were divided equally to both groups?
L16: ROS define
L20: “in vitro and vivo” preferably italics and also in before vivo
L20: “SDH” define
L35: chronic exposure of which subjects? Live animals, humans?
L63: “in vitro and in vivo” italics
L74: 24 piglets divided into 3 groups? Also weight range? Define also briefly the experimental groups.
L192: define BUN and CRE
Author Response
Dear Reviewer,
Thank you very much for taking the time to review this manuscript. Please see the detailed responses in the attachment.

Reviewer 2 Report
Some concerns are listed below:
How did you add vitamin E and phoxim to the diet and make sure the piglets could get the amount of vitamin E and phoxim you added? What is the age of animals used in the experiment? More details should be added to the materials and methods.
Did you check the feed intake for 3 groups? If you did, are there any differences?
You tested oxidative stress makers in the serum, to what extent did the oxidative stress occurred in the kidney contributes to the whole body oxidative stress? Is it the major target for oxidative stress? How about other organs, for example, liver?
Scale bars should be added to histology and TEM images.
Did you frequently see swollen mitochondria after exposure to phoxim? Can you quantify the size of mitochondria?
Moderate editing of English language required
Author Response

(The authors gave the same response as above.)

Reviewer 3 Report
Dear Authors
I hope this letter finds you well. I would like to begin by expressing my appreciation for your valuable contribution to the research through your manuscript titled “Vitamin E inhibited phoxim-induced renal oxidative stress and 2 mitochondrial apoptosis in vitro and in vivo," which I had the opportunity to review. I would like to offer my feedback and suggestions to further enhance the quality and impact of your manuscript.
Firstly, I must commend your efforts in conducting the study and compiling the results. However, I have concerns about the introductory section of your manuscript. While the topic's importance is briefly mentioned, the background information provided is insufficient to adequately contextualize the research problem. A comprehensive introduction should establish the relevance of the study by clearly outlining the existing gaps in knowledge and explaining how your research addresses these gaps.
Additionally, the manuscript lacks a thorough justification for the chosen sample size. Justifying the sample size is crucial as it directly influences the study's validity and the generalizability of the results. It would be beneficial to include a discussion of the statistical or methodological rationale behind the sample size determination. This will not only bolster the reliability of your findings but also enable readers to evaluate the study's robustness. Furthermore, is an ethical concern to use more than the minimum necessary number of animals
Moreover, one significant aspect that is missing from your manuscript is a detailed analysis of the feeding behavior of the treated piglets. Given that the treatment likely has an impact on their feeding patterns, a comprehensive exploration of this aspect is essential. Including data on feeding behavior, such as changes in feed intake, feeding frequency, and any observed abnormalities, will provide a more holistic understanding of the treatment's effects on the animals.
Lastly, the timing at which the exposure to phoxim starts to impair the animals is a critical factor in your study. This information is fundamental for interpreting the progression of the treatment's effects and its potential implications. I encourage you to include a clear timeline indicating when the animals were exposed to phoxim and how this exposure correlates with observed impairments or changes in behavior.
In conclusion, I believe that addressing these concerns will substantially improve the quality and impact of your manuscript. Your research has the potential to make significant contributions to the field, and ensuring a well-structured introduction, a robust sample size justification, thorough analysis of feeding behavior, and detailed information about the timing of impairments will greatly enhance its value.
Some minors suggestion are highlighted in the attached file
Sincerely

Author Response

(The authors gave the same response as above.)

Round 2
Reviewer 1 Report
Although the authors have attempted to answer and revise certain sections, still the overall outcome of the study is questionable for its novelty.
Reviewer 2 Report
Again scale bars should be added to histology and TEM images.
As you mentioned you have submitted some data to another journal. Please mentioned the results and indicating that data are not shown in the current manuscript.
Response 2: Phoxim exposure resulted in significantly reduced average daily feed intake (ADFI), which was presented in other submitted articles.
Is the food intake of Phoxim+vitamin E group comparable with control group? or whether you add vitamin E food intake is always decreased after phoxim exposure?
The food intake might be one of major reasons that leads to the differences in physiological changes which needs to be discussed.
The title needs to be revised as well. Which animals/cell model did you use in the current experiment should be clarified in the title.
Fine
Reviewer 3 Report
Dear Authors,
I would like to express my gratitude for your dedication and effort in addressing the revisions requested for your article. Your commitment to improving the quality of your work is commendable, and it is evident that you have made significant changes to the manuscript. However, there is one critical aspect that still requires clarification, specifically regarding the rationale behind the chosen sample size.
In your revised manuscript, you state that you used a sample size of 8 for each experimental group. While I appreciate your willingness to adapt to reviewers' comments, it is essential to provide a clear justification for this sample size, especially in experimental design studies like yours. In such research, the choice of sample size is pivotal, as it directly affects the validity and generalizability of your findings.
It widely adapted that in this type of study, the minimum number of animals in each group should be 5. However, you chose 8 animals per group. I would like to inquire about the reasoning behind this decision. Did you have any specific data or references from the literature that supported the use of 8 animals as opposed to the minimum recommended number of 5? It is crucial to demonstrate that your chosen sample size was not arbitrary but based on sound scientific reasoning or previous empirical evidence.
Additionally, I would appreciate it if you could elaborate on the statistical techniques or calculations you used to determine the sample size of 8. Were there any specific power calculations, effect sizes, or assumptions that guided your choice? Providing this information would help readers and fellow researchers understand your methodology better and evaluate the robustness of your experimental design.
In scientific research, transparency and rigor in sample size determination are fundamental to ensure the validity and reliability of study outcomes. Therefore, I kindly request that you revise your manuscript to include a detailed explanation of why you chose a sample size of 8 and whether you had any empirical or theoretical justifications for this decision. This information will strengthen your research and enhance the credibility of your findings.
I look forward to your response and the revised manuscript. Thank you for your attention to this important matter.
Sincerely,
Round 3
Reviewer 2 Report
All my questions have been addressed. Please make the scale bar more clear. In your submitted manuscript, not all scale bars are clearly presented in figures.
Reviewer 3 Report
Dear Authors,
I hope this letter finds you well. I have carefully reviewed your scientific article and would like to provide some constructive feedback regarding your choice of sample size. Specifically, I would like to address the rationale behind the selection of 24 animals instead of eg 12 or 36, and what implications this choice has on the study.
In your article, you mentioned that the unique contribution of this research lies in the justification of the sample size being 24 animals. It is crucial to elaborate on the reasons for this decision to enhance the understanding and credibility of your study.
Here are some key points to consider when explaining the significance of the chosen sample size:
Statistical Power: Discuss how a sample size of 24 provides adequate statistical power to detect meaningful effects or differences in your study. Explain whether power calculations were conducted and how they influenced your decision.
Experimental Design: Clarify whether your research involves complex experimental designs, subgroup analyses, or multiple outcome measures that necessitate a larger sample size to ensure the validity of your findings.
Resource Constraints: If resource limitations influenced your choice, be transparent about them. Discuss any practical considerations that guided your decision-making.
Previous Literature: Reference existing literature or similar studies that have used a comparable sample size to support your choice. Explain how your study builds upon or deviates from these precedents.
Ethical Considerations: Address any ethical considerations related to the use of animals in your research and how these considerations influenced the sample size decision.
Generalizability: Discuss how the sample size aligns with your research objectives and whether the results can be reasonably extrapolated to the broader population or context of interest.
In summary, a well-justified sample size is essential for the credibility and validity of scientific research. Providing a clear and comprehensive rationale for the use of 24 animals in your study will strengthen your article and enhance its contribution to the field.
I encourage you to revise and expand upon the explanation of your sample size choice, ensuring that readers can appreciate the significance of this decision in the context of your research.
Thank you for considering my feedback, and I look forward to seeing the revised version of your article.
Sincerely,
Round 4
Reviewer 3 Report
Dear Authors,
I hope this letter finds you well. I am writing to inform you that I have reviewed your manuscript. I would like to express my appreciation for the opportunity to review your work and provide my feedback.
I particularly appreciate your thorough review of the existing literature to justify the sample size used in your study. While it is essential for research to have a robust sample size justification, it is not uncommon in certain areas of research, to rely on existing literature as a basis for sample size determination. Given the nature of your research topic, I believe your approach is justified and aligns with the current practices in the field.
Your manuscript also provides valuable insights and contributes significantly to the ongoing discourse in the field. The results and discussions are well-supported by the literature, and your conclusions are sound based on the data presented.
In conclusion, I am pleased to accept your manuscript for publication.
Sincerely,
